# Drivers of vehicle-to-everything (V2X) adoption: A behavioral reasoning theory perspective

Ibrahim Arpaci[1,2,3]*, Mohammed A. Al-Sharafi[4], Moamin A. Mahmoud[5,6]

1 Management Information Systems, College of Business Administration, Gulf University for Science and Technology, Mishref, Kuwait, 2 Department of Software Engineering, Faculty of Engineering and Natural Sciences, Bandirma Onyedi Eylul University, Balikesir, Türkiye, 3 Department of Computer Science and Engineering, College of Informatics, Korea University, Seoul, Republic of Korea, 4 IRC for Finance and Digital Economy, KFUPM Business School, King Fahd University of Petroleum & Minerals, Dhahran, Saudi Arabia, 5 Institute of Informatics and Computing in Energy, Universiti Tenaga Nasional, Selangor, Malaysia, 6 College of Computing and Informatics, Universiti Tenaga Nasional, Selangor, Malaysia

* ibrahimarpaci@gmail.com

## Abstract

Vehicle-to-Everything (V2X) is a critical technology that enhances safety, improves traffic efficiency, and paves the way for future vehicles, such as autonomous cars, by enabling vehicles to communicate with each other and their environment. This study investigated the relationships among the reasons for and against adopting V2X technology, environmental values, attitudes, intentions, and green behavior. Accordingly, a research model was developed based on "Behavioral Reasoning Theory" and validated through PLS-SEM, using data from electric vehicle (EV) drivers. The findings indicated that environmental values positively predict the reasons for V2X adoption, which positively predict attitudes toward V2X adoption. In contrast, environmental values negatively predict the reasons against V2X adoption, which negatively predict attitudes toward V2X adoption. These insights are crucial for understanding the dynamics that shape individuals' attitudes toward V2X adoption and green behavior.

## 1. Introduction

The drive toward transitioning to electric vehicles (EVs) has gained significant momentum, evidenced by over 20 countries' mandates to sell EVs exclusively by 2030 [1]. This transition will substantially reduce tailpipe greenhouse gas emissions (2). However, concerns about the potential pressure on power systems due to increased electrical demand could lead to load congestion, frequency and voltage deviations, and prolonged power outages [2]. The simultaneous charging of many EVs, especially during peak electricity demand periods, may necessitate costly upgrades to grid capacity [3].

In Europe, electric car sales reached 1,774,000, a 48.3% increase in the first 10 months of 2023. The market share of electric cars also rose from last year's 11.7% to

Data availability statement: Data are publicly available at: https://github.com/iarpaci/myresearchdata

Funding: This work was supported by the Dato' Low Tuck Kwong International Energy Transition Grant under the project code of 202202002ETG. This work has been partially supported by Gulf University for Science and Technology (GUST) and the GEAR Research Center under project code ISG – Case #89. This work acknowledges the support of the GUST for covering the open access Article Processing Charge (APC). The funders had no role in study design, data collection and analysis, the decision to publish, or the preparation of the manuscript.

14.6% this year. During the same period, in Turkey, electric car sales reached 48,883, marking an 877% increase [4]. While the market share of electric cars in Europe, with significant incentives, grew by only 25% in 10 months, the growth rate in Turkey reached 427%. The market share of electric cars in Europe is 14.6%, whereas in Turkey, it has reached 6.5% [4]. The launch and increased usage of Turkey's domestic car, TOGG, along with the growing presence of charging stations, particularly in city centers, have contributed to a rise in demand for EVs in Turkey. In 2024, the number of electric car sales in Turkey is expected to surpass 100,000, with a corresponding market share of over 10% [4].

Innovations in EV technology are increasingly focusing on transforming EVs into bidirectional energy sources, using their onboard batteries as auxiliary power reservoirs [5]. This has led to the advancement of "Vehicle-to-Everything" (V2X) technology, where the power in EV batteries can be harnessed for diverse applications. These include powering devices (V2L), supplying energy to buildings (V2H/V2B), and supporting the electrical grid (V2G) [6]. Beyond mitigating peak loads, V2X technology has the potential to provide additional benefits at both the transmission and distribution levels. These benefits include frequency and voltage regulation and stabilizing renewable energy sources [7].

V2X involves using EV batteries to deliver energy services, extracting value from batteries when the EV is not in use [8]. These services optimize battery use through dynamic charging control, providing benefits to the grid system such as energy consumption management in buildings or homes and offering backup power [9]. Energy services are "the services enabled by dynamic control charging, leading to increased adaptable capacity in ancillary service markets and the wholesale sector" [10]. This adaptability proves advantageous for system operators and other stakeholders engaged in the technical management of the electric grid [10].

The development of 5G technology has made it feasible for vehicles to exchange information from the connected environment and predict future driving situations through "vehicle-to-everything" (V2X) technology [11,12]. The V2X technology encompasses various operational modes and electrical connections, including "vehicle-to-grid (V2G)," "vehicle-to-home (V2H)," "vehicle-to-vehicle (V2V)," "vehicle-to-building (V2B)," "vehicle-to-load (V2L)," and "vehicle-for-grid (V4G)" [13]. V2V enables data exchange between vehicles [14].

While much research has focused on V2G, it is essential to note that V2X encompasses a range of topologies, with V2G being just one example [8]. V2G technology utilizes EVs as distributed energy resources, enabling bidirectional power exchange with the grid, thereby enhancing grid reliability and efficiency [10]. Integrating crucial bi-directional power electronic converters, including DC-DC and AC-DC types [15], facilitates a dynamic interaction between EVs and the electrical grid. These converters increase power generation capacity by storing or releasing energy as needed and improving grid reliability, stability, and efficiency. This integrated system supports seamless energy transfers for both "Grid-to-Vehicle" (G2V) and "Vehicle-to-Grid" (V2G) functions, emphasizing the pivotal role of these converters in the V2G ecosystem [15].

Advancements in AI are paving the way for "Intelligent Transportation Systems" (ITS) [16]. Vehicles are becoming more intelligent, which enhances their ability to interact with the environment. "Cellular Vehicle-to-Everything (C-V2X)" is an advanced technology that allows for communication among pedestrians, vehicles, and infrastructure via cellular networks, improving road safety and traffic efficiency [17]. A V2X communication system is designed to address traffic efficiency, road safety, and energy efficiency [17]. Applications of V2X contribute to traffic management, "Plug-in Electric Vehicle" (PEV) charging optimization, and improved location-based services [18]. However, developing accurate traffic flow prediction algorithms remains challenging [19].

This study introduced a comprehensive model for predicting the adoption of V2X technology, with a specific focus on electric vehicle drivers. While numerous review papers exist that delve into various aspects of V2X, we argue that a comprehensive theoretical model is necessary to systematically elucidate the key factors that predict the adoption of V2X technology. The proposed model is designed to provide a thorough and integrated framework, enhancing our understanding of the adoption dynamics of V2X technology among electric vehicle drivers.

Given the complexities and transformative potential of V2X technology, it is critical to introduce a predictive model for its adoption [20]. This study aims to investigate the prevailing attitudes of EV drivers toward V2X technologies and to understand the underlying rationale behind these attitudes. Utilizing the "Behavioral Reasoning Theory" (BRT), this study aimed to identify the factors that predict EV drivers' intentions to adopt V2X and engage in environmentally friendly behavior. The goal is to identify key factors that influence their openness to V2X. Adopting a quantitative methodology, this exploratory research is well-suited for the early stage of V2X research. The proposed model is designed to assess the positive and negative factors that drive V2X adoption, including the influence of environmental values on decision making.

The following section introduces a comprehensive model pertinent to V2X technology and outlines the study hypotheses. Section 3 outlines the research methodology, encompassing sampling, data collection, analysis, and measurements. Section 4 examines the findings derived from the data analysis. Section 5 interprets the key conclusions, comparing them with those of earlier studies. The article concludes by considering the practical and theoretical implications of the findings, recognizing the limitations, and suggesting avenues for future research.

## 2. Theoretical framework and hypothesis formulation

### 2.1 Behavioral reasoning theory

Scholars leverage theory to predict and explain the relationships among various variables in a given behavior [21]. Numerous theories outline diverse determinants of human behavior, including the "Theory of Reasoned Action (TRA)" [22], the "Theory of Planned Behavior (TPB)" [23], and the "Theory of Explanation-Based Decision Making (TEDM)" [24]. While these behavioral theories have empowered practitioners and researchers to comprehend the consumers' decision-making processes in various contexts, they face several limitations. Scholars have questioned the ability to determine and predict consumer behaviors [25].

Theories concerning behavioral intention suggest that belief concepts, encompassing normative beliefs, behavioral beliefs, and control beliefs, serve as predictors for "subjective norms," "attitudes," and "perceived behavioral control" [26]. Despite offering valuable insights into context specific factors predicting behavior, belief concepts have received relatively limited attention [27]. Furthermore, behavioral theories have not systematically explored whether and how concepts related to "reason" offer unique insights into motivational mechanisms. Consequently, introducing the "Behavioral Reasoning Theory" establishes theoretical connections between individuals' beliefs, reasons, intentions, and actual behavior [20].

BRT is an innovative framework in the domain of marketing that extends foundational theories of technology acceptance, such as the "Theory of Planned Behavior" (TPB). It explains the relations between values, reasons (against and for), attitudes, and user behavior. BRT is particularly suitable for this study, which adopts V2X technology, because it focuses on the reasoning behind individuals' behaviors. It offers a structured approach to analyzing the multifaceted and often conflicting considerations influencing EV drivers' adoption decisions. The rationale for choosing BRT over other

adoption theories is its unique ability to include additional cognitive paths, considering both reasons against and for a behavior. This approach enhances the understanding of human behavior by capturing the actions and motivations behind them [28]. In the context of V2X adoption, drivers' motivations and hesitations are influenced by diverse factors, including environmental considerations, economic reasoning, and technological concerns.

By utilizing BRT, the study examines the specific thoughts and beliefs that lead to the rejection or acceptance of V2X technology. The theory also enables the study to assess the relative importance of these factors, which are crucial for developing effective interventions or policies to promote the acceptance of V2X. In the rapidly evolving field of intelligent transportation systems, where technological advances often outpace behavioral research, BRT offers a robust and adaptable framework. It helps us understand how potential users will likely receive new technologies like V2X. This theoretical grounding is expected to provide valuable, academically interesting insights and to be practically relevant to stakeholders seeking to promote sustainable and intelligent transportation solutions. The visual presentation of the theoretical model, as shown in Fig 1, encapsulates this comprehensive lens through which the EV drivers' decision-making process is viewed, confirming BRT as an ideal choice for investigating the complex phenomenon of V2X technology adoption.

## 2.2 Study hypotheses

**2.2.1 Green behavior.** Green behavior refers to "the range of activities individuals undertake to lessen their environmental impact and contribute to environmental sustainability" [29]. Green behavior requires using environmentally responsible driving techniques and participating in the environmental protection provided by V2X technology [2]. The term "behavioral intention," as defined by [23], measures an individual's readiness to engage in a specific behavior, indicating a predisposition to engage in certain actions in the future. Behavioral intention is crucial in this context as it represents the likelihood of individuals' commitment to the future use of V2X technology.

The extant literature highlights the significant influence of behavioral intentions on green behavior, pointing to a propensity for adopting environmentally friendly technologies [30,31]. Prior research indicated a positive impact of green consumption intention and green behavior [32]. Likewise, prior research demonstrated that green production intention significantly shapes green behavior [33]. Based on these insights, the hypothesis proposed is as follows:

H1. A positive correlation exists between the behavioral intention to adopt V2X and the adoption of green behavior.

**2.2.2 Attitude.** Attitude, which is shaped through analytical and deliberative evaluations, reflects an individual's overall tendency or orientation toward a behavior [22]. In this study, attitudes towards V2X technology include personal interest, perception of the technology as an innovative solution, recognition of its numerous advantages, and the belief that owning a V2X-equipped vehicle would bring happiness [34]. The relation between attitude and behavioral intention (BI) is well established in theoretical frameworks such as TPB, TRA, and TAM [35]. This research indicates that individuals' attitudes and motives significantly impact their behavioral intentions. Empirical research supports this theoretical premise. Supporting this, a recent study found that attitudes toward "electric vehicles" (EVs) positively affect consumers' behavioral intentions [36]. Similarly, another study indicated a positive association between attitude and intentions to adopt battery swap technology [37]. Drawing on these findings, we posit that favorable attitudes towards V2X technology will significantly impact the intention to engage in green behavior related to V2X. Therefore,

H2. A positive relationship exists between attitude and behavioral intention to adopt V2X.

**2.2.3 Reasons for.** In the context of BRT, reasons encompass positive and negative factors influencing the decision to engage in a particular behavior [20]. The theory posits that the strength of the reasons for behavior is correlated with the association between global motives and the actual performance of that behavior. Within the scope of V2X adoption, factors such as environmental concerns, economic benefits, and technophilic aspects are considered. The environmental benefits of V2X technology are manifold. These include reduced emissions and thus improved air quality, an eco-friendly driving experience, a reduced carbon footprint, promotion of renewable energy, and more efficient traffic flow [33]. These advantages highlight the growing awareness of ecological challenges and the importance of personal responsibility in

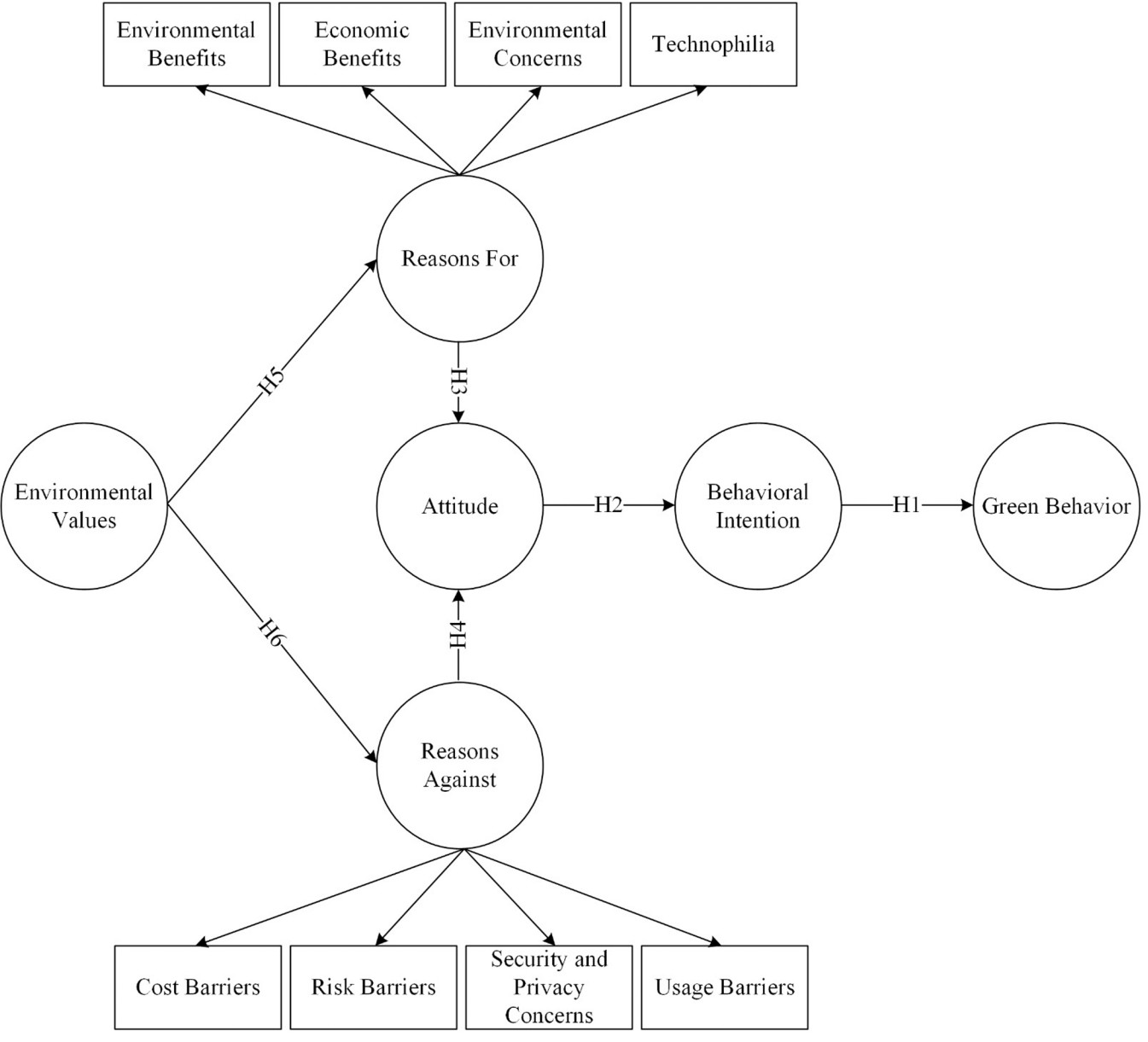

**Fig 1. Theoretical model.**

advancing sustainable practices [38]. Environmental concerns encompass worries, awareness, and responsibility for ecological issues [39]. V2X technology has the potential to positively contribute to sustainable living in harmony with nature, emphasizing the obligation to do more to protect depleting natural resources and individual responsibility for environmental protection [10].

Economic benefits are another strong incentive for V2X adoption. They can be listed as cost savings through lower insurance premiums, efficient routing, optimized driving for reduced energy use, fewer traffic delays, and longer component life [40]. The economic impacts of V2X, especially in terms of enhanced efficiency and bi-directional charging capabilities, are well-documented in the literature. Studies on V2G charging emphasize the importance of transparent

communication, financial compensation, and user-controlled reliability for system acceptance [41]. Additionally, V2G technology facilitates trading surplus and deficit energy, providing additional revenue and system-wide benefits [42,43]. This is supported by the findings of the Canary Islands and Latvia, where renewable energy storage through V2G reduces costs [44,45].

Technophilia also drives interest in V2X technology, characterized by a passion for technological innovation and a propensity to be an early adopter [46]. V2X's role in environmental sustainability is enhanced by its contribution to fuel efficiency and emission reduction. V2G technology enables incorporating renewable energy into electricity grids while helping to manage demand during periods of high usage [47], releasing excess EV battery energy to manage peak demand [48]. Consequently, V2X technology enhances the transportation experience by offering real-time information and promising a safer, more efficient, and interconnected transportation ecosystem. It is anticipated that individuals with reasons favoring V2X adoption, including enhanced efficiency, economic benefits, and environmental advantages, are likely to exhibit more positive attitudes toward the technology. Consequently,

H3. A positive relationship exists between the reasons for V2X adoption and individuals' attitudes toward the adoption.

**2.2.4 Reasons against.** While V2X communication offers promising advantages, its widespread adoption faces significant challenges. These challenges include risk barriers, cost barriers, usage barriers, and concerns about security and privacy [49]. Risk barriers encompass concerns about unreliable connections, potential inaccuracies in information exchange that can lead to vehicle coordination errors, and worries about the robustness of security measures against unauthorized access [50]. Interoperability challenges between manufacturers and service providers can complicate these risks [51]. There are also potential risks associated with the evolving nature of V2X technology and the need for ongoing updates. Cost barriers are also significant obstacles to adoption, with concerns about the high initial purchase price of V2X-enabled vehicles, potential daily operating costs, and higher maintenance and repair expenses [52]. Financial challenges are exacerbated by significant infrastructure investments to deploy communication networks and roadside charging units.

Usage barriers include concerns regarding learning to use V2X features in next-generation vehicles, the perceived complexity of integrating V2X into traffic systems, and the need for frequent updates and technical support [53]. Security concerns regarding V2X-equipped vehicles include unauthorized access, data security, privacy protection, risk of cyber-attack exposure, and doubts about the complete security of V2X communications against surveillance or hacking [54]. The exchange of sensitive information in V2X systems raises concerns over privacy and security, necessitating robust measures to safeguard against unauthorized access and malicious activities.

A previous study has identified additional factors that negatively affect the acceptance of related technologies such as V2G, including discomfort during participation, range anxiety, and concerns about battery degradation [41]. Overcoming these barriers requires building public trust, navigating complex regulations, and mitigating the environmental impact of infrastructure deployment. These considerations must be balanced to harness the full potential of V2X while ensuring its safety, security, and sustainability. Individuals with reasons against V2X adoption are likely to have less favorable attitudes toward the technology. Thus,

H4. A negative relationship exists between reasons against V2X adoption and individuals' attitudes toward the adoption.

**2.2.5 Environmental values.** Values are an individual's cognitive structures and subjective probability judgments that forecast appropriate future behavior [22]. These values are intimately connected to cognitive patterns that inform decision-making and subjective assessments [22]. Theories such as the "Theory of Explanation-Based Decision Making" (TEDM) and "Reasons Theory" [55] posit a strong connection between beliefs or values and reasons, as highlighted by [28]. In this study, environmental values are characterized by recognizing nature's intrinsic worth, striving for harmonious coexistence with the environment, actively protecting and preserving the environment, and understanding the critical importance of these efforts for sustainable future transportation [33]. These values are guiding principles, influencing perspectives and decision-making throughout the study.

Research into factors influencing interest in adopting EVs has revealed the interplay of behavioral, socio-demographic, economic, and technical elements. Financial savings, fuel economy, and environmental value influence attitudes toward EV adoption [56]. The ecological benefits of V2G technology, including its potential to address climate change and mitigate air pollution, are substantial. A study in Texas revealed a significant reduction in $CO_2$ emissions (59%) with the use of V2G in "Plug-in Electric Vehicles" [57]. A study modeling V2G adoption in San Francisco found significant energy cost savings with measurable $CO_2$ reductions and surprisingly fast payback periods [58]. Therefore, environmental values are expected to correlate positively with reasons for and against certain behaviors. We hypothesize a positive correlation between individuals who hold strong ecological values, such as respecting and adapting to nature, living in harmony with nature, and the conviction that actions must protect and preserve nature, and their support for V2X adoption. Conversely, individuals not prioritizing environmental values may exhibit higher skepticism or resistance to V2X adoption. Accordingly,

H5. A positive relationship exists between individuals who strongly prioritize environmental values and their endorsement of reasons for adopting V2X.

H6. A negative relationship exists between individuals who strongly emphasize environmental values and their inclination to oppose reasons against adopting V2X.

## 3. Research methodology

### 3.1 Sample and procedure

A total of 1,289 responses were received in this study. However, due to purposive sampling, the focus was on individuals with prior experience with EVs. Consequently, 74.24% of respondents who were not EV drivers were excluded. Among the participants, 332 reported owning or leasing an electric car. The majority (49.7%) used their EVs daily, followed by several times a week (23.5%), once a week (12.3%), and a few times a month (14.5%). The participant demographic consisted of 37.7% females (208 males and 124 females), with a mean age of 26.81, a range of 18–77 years, and a standard deviation of 9.156. Regarding environmental engagement, participants expressed high involvement, with 41% being very engaged, 33.4% somewhat engaged, 18.7% unsure, 5.4% not very engaged, and 1.5% not engaged.

Data were collected using Google Forms between September 15, 2023, and February 15, 2024. The survey link was distributed through social media platforms in Turkiye to reach a diverse pool of respondents. Convenience sampling was employed, and participants willingly volunteered for the study. This research was approved by the "Institutional Review Board" (IRB) of Bandirma Onyedi Eylül University (# 2023−7). The study followed the approved protocol without any deviations. Before participation, all participants were briefed on the research's objectives and provided with a clear explanation, ensuring transparency regarding the use of their data, which would be exclusively for research purposes. Participants obtained written consent through the online survey platform, where they explicitly agreed to participate after reviewing the study details and information.

### 3.2 Measurements

The study employed a comprehensive measurement approach to capture the varied perceptions and barriers associated with adopting V2X technology. The measurement includes 47 items designed to evaluate 12 distinct constructs, ensuring a precise assessment of both dependent and independent variables as outlined in the proposed research model. The measurement is divided into two parts: Part 1 covers demographic variables, and Part 2 consists of the scale items. Demographic variables encompass gender, age, education level, driving frequency, vehicle ownership, electric vehicle ownership, and environmental engagement. The scale items are evaluated using a "five-point Likert-type scale" for responses. The scale items measuring each variable were adapted from prior studies. The measurement items and associated references are presented in S1 Appendix.

## 3.3 Data analysis

The study employed "Partial Least Squares Structural Equation Modeling" (PLS-SEM), implemented via SmartPLS-4, to assess the hypothesized relationships in our study. Several key factors influenced the decision to use PLS-SEM for analysis. First, the study aims to test a theoretical model from a prediction perspective, and the PLS-SEM approach is more suitable for this purpose. Moreover, the proposed model is complex as it encompasses numerous variables. Additionally, the path model seems to be even more compatible with the qualities of PLS-SEM as it includes formally measured constructs.

One crucial consideration in opting for PLS-SEM is the limitation imposed by a small population, which inherently constrains the available sample size [59]. This constraint highlights the necessity for a robust analytical approach that can effectively address the complexities of the structural model and the challenges posed by a limited sample size. Consequently, PLS-SEM is a suitable and strategic choice for this research context.

## 4. Results

### 4.1 Common method bias

"Common method bias (CMB)" refers to measurement errors that result from methodological flaws. For instance, using a single instrument to measure all items can lead to CMB. This research employed Harman's one-factor test to evaluate CMB [60]. This approach involved unrotated "exploratory factor analysis" using 47 items loaded on a latent construct. According to the findings, the single latent construct explains only 44.55% of the average variance. Since this value is below the recommended limit of 50%, CMB is not a serious problem for this study.

### 4.2 Measurement model

Before testing the relationships in the structural model, it is necessary to evaluate the measurement model. Thus, reliability, discriminant, and convergent validity were initially assessed [61,62]. Several analyses were conducted to determine reliability, including "Cronbach's Alpha" (CA) and "Composite Reliability" (CR). The reliability tests for each variable are detailed in Table 1. The reliability analysis results revealed that Cronbach's alpha value for each construct was above 0.70, showing adequate internal consistency reliability for the scale.

Table 1 presents the reliability and validity assessment of the constructs that showed strong psychometric properties across all measures. Each construct contained several items with factor loadings exceeding the recommended threshold of 0.70, indicating robust item-structure relationships [61,63]. High CA and CR scores, above 0.70, confirmed reliability, reflecting excellent internal consistency. Convergent validity is supported by "Average Variance Extracted" (AVE) values higher than 0.5, with EV and RB showing strong AVE scores of 0.758 and 0.742, respectively. Although EB has a slightly lower AVE (0.599), it remains within acceptable limits [61]. Overall, the results validate the measurement model, confirming that all constructs are reliably and accurately measured and making them suitable for further structural analysis.

Discriminant validity was assessed using heterotrait-monotrait (HTMT) ratios [64]. Table 2 shows that all HTMT values were below the 0.90 threshold, thus providing satisfactory discriminant validity for the measurement model [65].

### 4.3 Assessment of formative second-order constructs

The concept of multicollinearity in formative indicators is assessed using the "variance inflation factor" (VIF). For an indicator to be considered acceptable, it must meet specific criteria: A *p*-value below 0.05, loadings above 0.50, and a VIF value below 5. The results align with these thresholds, indicating that the weights of the overall indicators are significant. VIF scores ranged between 2.188 and 3.760, indicating that the degree of multicollinearity among the predictor variables is well below the commonly considered threshold of 5.0 proposed by [61,66]. The bootstrapping analysis results, presented in Table 3, uncovered a significant relationship between formative second-order constructs: Economic Benefits,

**Table 1. Measurement model assessment.**

| Constructs | Items | Loadings | CA | CR | AVE |
|---|---|---|---|---|---|
| Attitudes | At37 | 0.799 | 0.815 | 0.878 | 0.643 |
| | At38 | 0.816 | | | |
| | At39 | 0.785 | | | |
| | At40 | 0.805 | | | |
| Behavioral Intention | BI58 | 0.844 | 0.82 | 0.893 | 0.735 |
| | BI59 | 0.868 | | | |
| | BI60 | 0.86 | | | |
| Cost Barrier | CB26 | 0.79 | 0.743 | 0.854 | 0.661 |
| | CB27 | 0.837 | | | |
| | CB28 | 0.811 | | | |
| Environmental Benefits | EBe5 | 0.841 | 0.895 | 0.923 | 0.705 |
| | EBe6 | 0.832 | | | |
| | EBe7 | 0.839 | | | |
| | EBe8 | 0.843 | | | |
| | EBe9 | 0.843 | | | |
| Environmental Concerns | ECo15 | 0.826 | 0.816 | 0.879 | 0.646 |
| | ECo16 | 0.860 | | | |
| | ECo17 | 0.795 | | | |
| | ECo18 | 0.728 | | | |
| Environmental Values | EV1 | 0.881 | 0.893 | 0.926 | 0.758 |
| | EV2 | 0.888 | | | |
| | EV3 | 0.838 | | | |
| | EV4 | 0.874 | | | |
| Economic Benefits | EcB10 | 0.769 | 0.833 | 0.882 | 0.599 |
| | EcB11 | 0.792 | | | |
| | EcB12 | 0.747 | | | |
| | EcB13 | 0.762 | | | |
| | EcB14 | 0.799 | | | |
| Green Behavior | GB41 | 0.809 | 0.835 | 0.89 | 0.669 |
| | GB42 | 0.848 | | | |
| | GB43 | 0.800 | | | |
| | GB44 | 0.814 | | | |
| Risk Barrier | RB23 | 0.851 | 0.826 | 0.896 | 0.742 |
| | RB24 | 0.858 | | | |
| | RB25 | 0.874 | | | |
| Security and Privacy Concerns | SPC32 | 0.847 | 0.869 | 0.905 | 0.656 |
| | SPC33 | 0.779 | | | |
| | SPC34 | 0.847 | | | |
| | SPC35 | 0.791 | | | |
| | SPC36 | 0.784 | | | |
| Technophilia | T19 | 0.849 | 0.846 | 0.896 | 0.684 |
| | T20 | 0.825 | | | |
| | T21 | 0.826 | | | |
| | T22 | 0.808 | | | |

*(Continued)*

**Table 1.** (Continued)

| Constructs | Items | Loadings | CA | CR | AVE |
|---|---|---|---|---|---|
| Usage Barrier | UB29 | 0.845 | 0.817 | 0.891 | 0.733 |
| | UB30 | 0.869 | | | |
| | UB31 | 0.853 | | | |

**Table 2. HTMT results.**

| Construct | 1 | 2 | 3 | 4 | 5 | 6 | 7 | 8 | 9 | 10 | 11 |
|---|---|---|---|---|---|---|---|---|---|---|---|
| 1. Attitudes | | | | | | | | | | | |
| 2. Behavioral Intention | .841 | | | | | | | | | | |
| 3. Cost Barrier | .577 | .547 | | | | | | | | | |
| 4. Economic Benefits | .785 | .726 | .547 | | | | | | | | |
| 5. Environmental Benefits | .78 | .707 | .552 | .883 | | | | | | | |
| 6. Environmental Concerns | .773 | .697 | .541 | .861 | .895 | | | | | | |
| 7. Environmental Values | .739 | .704 | .552 | .767 | .796 | .737 | | | | | |
| 8. Green Behavior | .861 | .825 | .571 | .759 | .794 | .763 | .743 | | | | |
| 9. Risk Barrier | .601 | .536 | .874 | .546 | .541 | .482 | .51 | .584 | | | |
| 1. Security and Privacy Concerns | .626 | .493 | .716 | .476 | .505 | .487 | .449 | .586 | .835 | | |
| 11. Technophilia | .89 | .742 | .663 | .831 | .872 | .805 | .779 | .806 | .602 | .617 | |
| 12. Usage Barrier | .506 | .44 | .879 | .449 | .477 | .401 | .42 | .466 | .894 | .833 | .554 |

**Table 3. Higher-order construct validity.**

| HOC | LOCs | B | T | P | VIF |
|---|---|---|---|---|---|
| Reasons For | Economic Benefits | 0.224 | 3.401 | 0.001 | 2.824 |
| | Environmental Benefits | 0.244 | 2.967 | 0.003 | 3.760 |
| | Environmental Concerns | 0.163 | 2.307 | 0.021 | 2.746 |
| | Technophilia | 0.482 | 7.962 | 0.000 | 2.608 |
| Reasons Against | Risk Barrier | 0.373 | 2.644 | 0.008 | 2.891 |
| | Security and Privacy Concerns | 0.495 | 3.993 | 0.000 | 2.356 |
| | Cost Barrier | 0.442 | 3.686 | 0.000 | 2.188 |
| | Usage Barrier | −0.192 | 1.980 | 0.046 | 2.858 |

Environmental Benefits, Environmental Concerns, Technophilia, and Reasons For ($p < 0.001$). Risk Barriers, Security and Privacy Concerns, and Cost Barriers also exhibited a significant relationship with Reasons Against.

### 4.4 Hypothesis testing results

Before initiating hypothesis testing, the adequacy of the model fit was assessed by the following fit indices: "Standardized Root Mean Square Residual (SRMR)" = 0.043, Chi-square = 591.338, and "Normed Fit Index (NFI)" = 0.900. In line with the thresholds recommended by [67], an SRMR value lower than 0.08 indicates a good fit between the research model and the data. Table 4 indicates F-square values along with the results of the hypothesis testing.

The PLS-SEM bootstrapping analysis, with 5,000 resamples, indicated a positive and significant relationship between green behavior and behavioral intentions ($\beta = 0.684$, $p < 0.001$), supporting H1. Moreover, the findings revealed a positive and significant relationship between attitude and behavioral intentions ($\beta = 0.690$, $p < 0.001$), supporting H2. Furthermore,

**Table 4. Hypothesis testing.**

| H | Path | β | T | P values | F-square | Support |
|---|------|---|---|----------|----------|---------|
| H1 | Behavioral Intention → Green Behavior | 0.684 | 18.236 | 0.000* | 0.879 | Yes |
| H2 | Attitude → Behavioral Intention | 0.69 | 20.769 | 0.000* | 0.91 | Yes |
| H3 | Reasons For → Attitude | 0.67 | 14.385 | 0.000* | 0.043 | Yes |
| H4 | Reasons Against → Attitude | −0.163 | 3.136 | 0.002** | 0.73 | Yes |
| H5 | Environmental Values → Reasons For | 0.752 | 26.311 | 0.000* | 1.302 | Yes |
| H6 | Environmental Values → Reasons Against | −0.489 | 10.612 | 0.000* | 0.314 | Yes |

$* p < 0.001$, $** p < 0.01$

individuals' attitudes toward V2X adoption were positively predicted by the reasons for adoption ($β = 0.670$, $p < 0.001$), while attitudes against V2X adoption were negatively predicted by the reasons against adoption ($β = −0.163$, $p < 0.01$). Thereby, H3 and H4 were confirmed. Environmental values positively predicted the reasons for V2X adoption ($β = 0.752$, $p < 0.001$), confirming H5. Furthermore, environmental values negatively predicted the reasons against V2X adoption ($β = −0.489$, $p < 0.001$), thus supporting H6. As presented in Fig 2, the proposed paths accounted for 0.239, 0.566, 0.606, 0.476, and 0.468 variances in reasons against, for, attitude, behavioral intention, and green behavior, respectively.

## 5. Discussion

This study developed a theoretical model that identifies the key factors predicting the adoption of "Vehicle-to-Everything" (V2X) technology among "electric vehicle" (EV) drivers. Accordingly, a theoretical model based on "Behavioral Reasoning Theory" (BRT) is proposed and evaluated using a PLS-SEM approach. BRT offers advantages over established technology adoption models such as TAM and UTAUT. First, it uniquely accommodates positive and negative behavioral reasoning in a unified framework, effectively addressing the attitude-behavior paradox prevalent in sustainable technology adoption. Second, the empirical results indicate that BRT provides significantly more explanatory power for attitudes, accounting for 61% of the attitude variance.

The PLS-SEM results indicated that the factors included in the research model significantly impact EV drivers' attitudes towards adopting V2X technology. The first hypothesis proposed a positive and significant relationship between behavioral intention and green behavior. This result revealed that behavioral intention is crucial in predicting green behavior. The results suggested that individuals' intention to adopt V2X technology significantly predicts their participation in environmentally friendly green behaviors. The result aligns with findings from earlier research, indicating a significant association between individuals' willingness to adopt V2X technology and their participation in environmentally conscious behaviors [30–32,68]. This is a crucial finding, highlighting the role of V2X technology as a technological advancement and a driving force for sustainable practices among its users.

The second hypothesis focused on the relationship between attitudes and behavioral intentions, and the findings revealed a strong and positive association between the variables. This result suggests that EV drivers' attitudes towards V2X technology significantly shape their intentions to engage in environmentally friendly actions. A positive attitude towards V2X technology can be described as a belief in its potential to improve driving efficiency, contribute to environmental sustainability, or provide personal convenience. A driver with a positive attitude towards environmental protection is more likely to use an electric vehicle in a way that minimizes carbon emissions. This finding aligns with prior research indicating that individuals' attitudes toward V2X technology substantially impact their behavioral intentions to adopt it [36,37].

Hypothesis 3 investigated the relationship between motivations to adopt V2X technology and individuals' attitudes towards adopting this technology. The results revealed a positive and significant relationship between reasons for

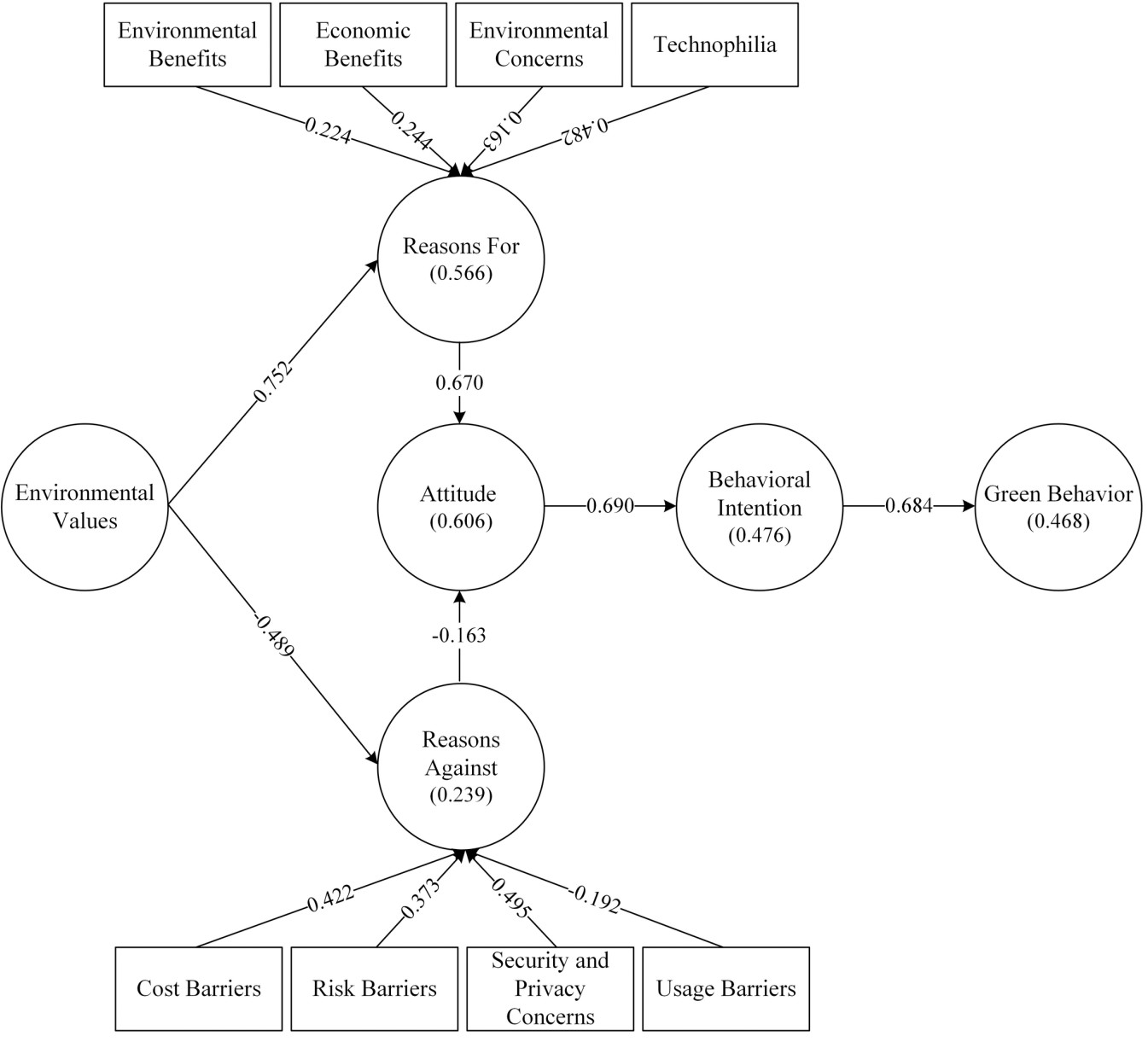

**Fig 2. Hypothesis testing results.**

adoption and behavioral intentions. The findings confirm that the reasons behind individuals' decisions to adopt V2X technology, including technophilia, environmental concerns, and the perceived benefits of both environmental and economic advantages, significantly shape their attitudes toward adopting this technology. This finding aligns with prior studies showing a positive and significant relationship between the reasons for a behavior and the actual behavior [20]. This finding also supports previous arguments regarding the environmental benefits identified in the existing literature, such as improving air quality, reducing emissions, promoting renewable energy, and efficient traffic flow [33]. This result emphasizes the importance of effectively communicating the benefits of V2X technology. When individuals realize tangible benefits, such as increased safety or reduced traffic congestion, their attitudes become more positive.

Hypothesis 4 investigated the negative relationship between the reasons against adopting V2X technology and individuals' attitudes toward its adoption. The results strongly support this hypothesis. As expected, this finding suggests that concerns such as risk barriers, usage barriers, cost barriers, and concerns about security and privacy negatively affect attitudes toward V2X technology. This result aligns with prior research conducted by [50,53,54], and [41], which also highlighted similar concerns impacting technology adoption. This outcome underscores the importance of positively addressing these negative perceptions to shape public attitudes toward V2X technology. It suggests that the more individuals are concerned about the risks, costs, usage difficulties, and privacy/security issues associated with V2X technology, the less favorable their attitudes toward its adoption are. Strategies to address these concerns are vital for fostering a more positive attitude towards V2X adoption. These might include implementing rigorous safety measures to alleviate risk concerns, transparent data handling, and privacy policies to address security and privacy worries, as well as simplifying the technology's usage to overcome usability barriers. Public awareness events that educate potential users about the advantages and safety features of V2X technology could also play a key role in altering public perception. These findings hold significant implications for policymakers, automotive manufacturers, and technology developers. They highlight the need for a multi-faceted approach to addressing the public's concerns regarding V2X technology. Future research could focus on developing effective communication strategies to mitigate these barriers, exploring the aspects of V2X technology that are most concerning to the public, and investigating the potential of various interventions to shift public attitudes positively.

This result highlights the importance of addressing these negative perceptions to shape public opinion and attitudes towards V2X technology positively. Accordingly, the more individuals are concerned about the risks, costs, usage challenges, and privacy and security issues associated with V2X technology, the less positive their attitudes toward its adoption. Therefore, addressing these concerns is crucial in encouraging a more positive attitude towards V2X adoption. These strategies could include implementing stringent security measures to alleviate risk concerns, transparent data handling, and privacy policies to address security and privacy concerns, as well as simplifying the use of the technology to overcome usability barriers. Public awareness events that educate potential users about the benefits and security features of V2X technology can also play a crucial role in changing public perception.

The findings have important implications for manufacturers, policymakers, and technology developers. They underscore the necessity for a multifaceted approach to addressing public concerns regarding V2X technology. To mitigate these barriers, stakeholders must develop effective communication strategies, explore the aspects of V2X technology that most concern the public, and ensure that various interventions positively impact public attitudes.

Hypothesis 5 proposed a positive association between environmental values and the reasons for V2X adoption. The results indicated a strong positive relationship between ecological values and reasons for adopting V2X. This suggests that individuals prioritizing environmental concerns are more likely to perceive positive reasons to engage in environmentally friendly behaviors associated with V2X technology. Those with strong ecological beliefs tend to recognize environmental benefits as key motivations for adopting V2X technology.

Hypothesis 6 proposed a negatively significant relationship between environmental values and reasons against V2X adoption. The findings confirmed the negative relationship between environmental values and the reasons against V2X adoption. This implies strong environmental values may reduce skepticism or resistance to V2X adoption. Based on the theory of reasons, this concept suggests that nurturing environmental values can be critical in overcoming perceived negativity or challenges associated with V2X technology [55]. This negative correlation is crucial in understanding the barriers to V2X adoption, implying that promoting environmental values can help overcome the perceived disadvantages or challenges associated with V2X technology. Highlighting the ecological advantages of V2X technology and aligning them with consumers' ecological values could be a strategic approach to reduce resistance.

The findings underscore the critical role of environmental values in shaping the reasons for and against adopting V2X. These findings have important implications for marketing strategies and policy development. Campaigns and policies that emphasize environmental benefits can effectively encourage the adoption of V2X technology, especially among

environmentally sensitive segments of the population. The findings are supported by empirical research in environmental behavior and technology adoption. The results underscore the significant impact of ecological values on shaping attitudes toward V2X technology and have important implications for marketing strategies and policy development. Tailoring campaigns and policies to highlight the environmental benefits of V2X technology can be particularly effective in promoting adoption among environmentally conscious individuals.

## 6. Conclusions

### 6.1 Theoretical implications

The findings have significant implications in the context of V2X technology adoption and substantially contribute to the existing literature. First, it extends the existing literature on technology adoption by applying the "Behavioral Reasoning Theory" (BRT) to the context of V2X technology. This theoretical framework provides a deeper understanding of the motivations underlying individuals' decisions to adopt new technologies. In particular, the study emphasizes the key role of environmental values, attitudes, and behavioral intention in predicting V2X adoption. The positive relation between environmental values and adoption motives and the influence of attitudes on behavioral intentions provides valuable insights into the motivational aspects of technology adoption. This research also enhances the understanding of how reasons against and for technology adoption influence consumer attitudes, thereby advancing the theoretical framework of BRT in the context of intelligent transportation systems.

Second, the proposed research model and this study's findings focus on the evolving landscape of transportation and connectivity, particularly in the context of smart cities and urban mobility. By examining consumer behavior and perceptions of V2X technology in a rapidly evolving urban environment, this study may contribute to the growing body of literature on smart city technologies and their impact on transportation. This contribution is particularly valuable as insights into specific technologies, such as V2X communication, are often missing in the literature on smart cities.

Third, the findings can enhance understanding of the factors that motivate and inhibit individuals' positive attitudes toward adopting and using V2X technology. It can fill an essential gap in prior research that has predominantly focused on adopting electric vehicles while neglecting the unique dynamics of V2X technology. This research focuses on V2X technology and offers new insights into the factors influencing consumer attitudes and behavioral intentions within an intelligent transportation system. Furthermore, this study differs from previous research by emphasizing the role of individual perceptions of safety and traffic efficiency in shaping attitudes toward V2X technology. While previous studies often focused on technological aspects, this research underlines the importance of user-centered factors. This shift in focus opens new avenues for future research and emphasizes the need to consider user perspectives and concerns when examining the adoption and use of V2X technology.

### 6.2 Practical implications

The findings can guide policymakers, automotive industry stakeholders, and marketers. Understanding the factors influencing V2X adoption can help develop targeted strategies to promote this technology. For example, emphasizing the environmental and economic benefits of V2X in marketing and education campaigns can positively influence consumer attitudes and adoption rates. The study also shows that addressing potential barriers to adoption, such as cost and security concerns, is crucial. For policymakers, the findings underscore the importance of establishing supportive regulatory frameworks and incentives that align with potential adopters' economic and environmental values. Furthermore, the automotive industry can leverage these insights to design and market V2X technologies that align with consumer values and attitudes.

The study's practical implications offer guidance for promoting the adoption of V2X technology through tailored marketing, addressing adoption barriers, creating supportive regulatory frameworks, user-centered design, industry collaboration, and environmental messaging. These strategies can help integrate V2X technology successfully into intelligent transportation systems.

## 6.3 Limitations and future research

While the study provides valuable insights, it is not without some limitations. Firstly, the findings of this study are based on a specific sample and context, which limits their generalizability. It is essential to recognize that attitudes and adoption factors may vary across regions, cultures, and demographics. Future work should confirm these findings in different cultural contexts, particularly in areas with varying levels of V2X infrastructure maturity (e.g., Europe, East Asia, or North America), to assess the universality of the BRT mechanisms identified in this study.

Secondly, this study utilizes cross-sectional data, providing a snapshot of attitudes and intentions at a specific time. Future studies should prioritize larger, more diverse samples of EV drivers, including sampling stratified across regions and demographics, to validate and extend our findings. Longitudinal designs could also track the attitudes and behavioral changes in V2X adoption. Furthermore, investigating cross-cultural differences in attitudes toward V2X adoption can provide potentially valuable insights into the role of cultural values and norms in predicting adoption decisions.

Thirdly, while BRT effectively captures reasoning processes, complementary theories can consider specific adoption factors such as habituation or social norms. Expanding the variables considered could provide a nuanced insight into adoption dynamics. Finally, the effectiveness of the instruments used in this study could be further improved. Future research could use more robust measurement tools and experimental designs to strengthen causal inferences. Additionally, qualitative research methodologies such as focus group interviews can be used to gain a deep insight into the reasons behind specific attitudes and perceptions towards V2X technology. Qualitative data can provide rich insights into the user experience.

## Supporting information

**S1 Appendix. Measurements.**
(DOCX)

## Author contributions

**Conceptualization:** Ibrahim Arpaci, Mohammed A. Al-Sharafi, Moamin A. Mahmoud.

**Data curation:** Mohammed A. Al-Sharafi.

**Funding acquisition:** Moamin A. Mahmoud.

**Investigation:** Ibrahim Arpaci.

**Methodology:** Ibrahim Arpaci.

**Resources:** Ibrahim Arpaci.

**Software:** Ibrahim Arpaci.

**Supervision:** Moamin A. Mahmoud.

**Validation:** Moamin A. Mahmoud.

**Writing – original draft:** Ibrahim Arpaci, Mohammed A. Al-Sharafi, Moamin A. Mahmoud.

**Writing – review & editing:** Ibrahim Arpaci, Mohammed A. Al-Sharafi, Moamin A. Mahmoud.

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
