## [Decision Letter · Decision Letter 0]

Dear Dr.  Arpaci,

Thank you for submitting your manuscript to PLOS ONE. After careful consideration, we feel that it has merit but does not fully meet PLOS ONE’s publication criteria as it currently stands. Therefore, we invite you to submit a revised version of the manuscript that addresses the points raised during the review process.

This study is overall well-organized. But revisions are still needed before qualifying for this journal publication.

We look forward to receiving your revised manuscript.

Kind regards,

Lei Zhang, PhD

Academic Editor

PLOS ONE

Journal Requirements:

 “This work was supported by the Dato’ Low Tuck Kwong International Energy Transition Grant under the project code of 202202002ETG.”

Additional Editor Comments:

This study is generally well organized. But there are still some places that need further revisions or clarifications before qualifying for this journal publication.

Reviewers' comments:

Reviewer's Responses to Questions

**Comments to the Author**

1. Is the manuscript technically sound, and do the data support the conclusions?

Reviewer #1: Yes

Reviewer #2: Yes

2. Has the statistical analysis been performed appropriately and rigorously?

Reviewer #1: Yes

Reviewer #2: No

3. Have the authors made all data underlying the findings in their manuscript fully available?

Reviewer #1: Yes

Reviewer #2: No

4. Is the manuscript presented in an intelligible fashion and written in standard English?

Reviewer #1: Yes

Reviewer #2: Yes

Reviewer #1: This article, based on Behavioral Reasoning Theory (BRT), explores the impact of environmental values, attitudes, behavioral intentions, and green behavior on V2X technology adoption, filling a gap in user behavior research in the V2X field. The research design is rigorous, using PLS-SEM to validate hypotheses, and the data collection and analysis processes are transparent. The results reveal the mechanism by which environmental values influence attitudes through dual paths of "reasons for and against," demonstrating theoretical innovation.

Reviewer #2: 1. The study samples are mainly concentrated in Türkiye, which may lead to regional deviation of results, limiting the universality and applicability of the study. Suggest expanding the sample size to include respondents from different countries and cultural backgrounds to enhance the generalizability and external validity of the study.

2. The study collected 1289 responses, of which only 332 were from EV drivers. The sample size may not be sufficient to fully represent the entire EV driver population, leading to regional bias in the results. Suggest increasing the sample size, especially among the target population (EV drivers), to improve the robustness and reliability of statistical analysis.

3. The use of convenience sampling and online surveys in research may introduce selection bias. It is recommended to adopt stricter random sampling methods and combine them with offline surveys to improve sample representativeness and reduce selection bias.

4. Research based on cross-sectional data cannot reveal the dynamic changes of attitudes and behaviors over time. Suggest conducting longitudinal research to track the attitudes and behavioral changes of EV drivers towards V2X technology, in order to provide a deeper understanding.

5. Although the model adopts Behavioral Reasoning Theory (BRT), it may not cover all factors that affect the adoption of V2X technology. It is suggested to consider incorporating other relevant theories (such as technology acceptance models, social impact theories, etc.) into the research framework to construct a more comprehensive theoretical model.

6. The literature review section may not cover all important research in the relevant fields. Suggest citing "Interacting multiple model-based ETUKF for efficient state estimation of connected vehicles with V2XV communication" and "Research on multi-lane energy-saving driving strategy of connected electric vehicle based on vehicle speed prediction" to expand the scope of the literature review and provide more comprehensive background support on V2X related technology.

7. As for the theoretical model developed based on "behavioral reasoning theory" proposed in this paper, the experimental part lacks the comparison between this method and other SOTA methods. It is suggested to use SOTA methods in related research fields as the baseline to prove the progressiveness of the proposed method. 

8. For the theoretical model developed based on the "behavioral reasoning theory" proposed in this paper, some other quantitative indicators should be selected or additionally involved to reflect the progressiveness of the method proposed in this paper.

**Do you want your identity to be public for this peer review?** For information about this choice, including consent withdrawal, please see our Privacy Policy

Reviewer #1: No

Reviewer #2: No

---

## [Author Response · Author response to Decision Letter 1]

6 May 2025

We addressed all comments in a separate document.

---

## [Decision Letter · Decision Letter 1]

Drivers of Vehicle-to-Everything (V2X) Adoption: A Behavioral Reasoning Theory Perspective

PONE-D-25-07382R1

Dear Dr. Ibrahim,

We’re pleased to inform you that your manuscript has been judged scientifically suitable for publication and will be formally accepted for publication once it meets all outstanding technical requirements.

Kind regards,

Lei Zhang, PhD

Academic Editor

PLOS ONE

Additional Editor Comments (optional):

The revised submission is publishable in current form.

Reviewers' comments:

Reviewer's Responses to Questions

**Comments to the Author**

Reviewer #1: All comments have been addressed

Reviewer #2: All comments have been addressed

2. Is the manuscript technically sound, and do the data support the conclusions?

Reviewer #1: Yes

Reviewer #2: Yes

3. Has the statistical analysis been performed appropriately and rigorously?

Reviewer #1: Yes

Reviewer #2: Yes

4. Have the authors made all data underlying the findings in their manuscript fully available?

Reviewer #1: Yes

Reviewer #2: No

5. Is the manuscript presented in an intelligible fashion and written in standard English?

Reviewer #1: Yes

Reviewer #2: Yes

Reviewer #1: This study, grounded in Behavioral Reasoning Theory (BRT), investigates the influence of environmental values, attitudes, behavioral intentions, and green behavior on the adoption of Vehicle-to-Everything (V2X) technology, addressing a gap in user behavior research within this field. Using Partial Least Squares Structural Equation Modeling (PLS-SEM), the study validates hypotheses and demonstrates that environmental values significantly shape user attitudes through dual pathways of "reasons for and against" adoption, showcasing theoretical innovation. While the research design is rigorous and the data analysis transparent, limitations include a geographically restricted sample (primarily from Turkey), a relatively small sample size (only 332 electric vehicle drivers), and reliance on cross-sectional data that cannot capture dynamic behavioral changes. Future research could enhance generalizability by expanding the sample across diverse regions, incorporating longitudinal designs, and integrating additional theoretical frameworks (e.g., Technology Acceptance Model) for a more comprehensive understanding.

Reviewer #2: All comments has been addressed. After modification, the conclusion of the article is more rigorous, the structure of the article is more reasonable, and it has a certain degree of innovation.

**Do you want your identity to be public for this peer review?** For information about this choice, including consent withdrawal, please see our Privacy Policy

Reviewer #1: No

Reviewer #2: No

---

## [Editor Report · Acceptance letter]

PONE-D-25-07382R1

PLOS ONE

Dear Dr. Arpaci,

I'm pleased to inform you that your manuscript has been deemed suitable for publication in PLOS ONE. Congratulations! Your manuscript is now being handed over to our production team.

Kind regards,

on behalf of

Dr. Lei Zhang

Academic Editor

PLOS ONE